# Transformation of the Microstructure of Fe-Cr Steel during Its Production

Andrés Núñez [1,2,*], Irene Collado [1,3], Juan F. Almagro [1] and David L. Sales [2]

1 Laboratory & Research Section, Technical Department, Acerinox Europa S.A.U., 11379 Palmones, Spain; irene.collado@acerinox.com (I.C.); juan.almagro@acerinox.com (J.F.A.)

2 INNANOMAT Group, Department of Materials Science, Metallurgical Engineering and Inorganic Chemistry, Higher Polytechnic School, University of Cádiz, 11202 Algeciras, Spain; david.sales@uca.es

3 LABCYP Group, Department of Materials Science, Metallurgical Engineering and Inorganic Chemistry, School of Engineering, University of Cádiz, 11519 Puerto Real, Spain

* Correspondence: andres.nunez@acerinox.com

**Abstract:** EN 1.4016 stainless steels combine good corrosion resistance with good formability and ductility. As such, their most popular applications are related to sheet forming. During re-crystallisation of Fe-Cr steels, deviations from the desired $\gamma$-fibre (gamma fibre, <111>||ND) texture promote a decrease in deep drawability. Additionally, $\alpha$-fibre (alpha fibre, <110>||RD) has been found to be damaging to formability. In this study, an EN 1.4016 basic material and a modified one with optimised settings as regards to chemical composition and manufacturing process, in order to improve the formability properties, are characterised. The phase diagram, microstructure, Lankford coefficients and Electron Backscatter Diffraction (EBSD) (results confirm the evolution of texture during the processing of ferritic stainless steel. Texture is analysed by the interpretation of Orientation Distribution Function (ODF), using orientation density results for each sample obtained in the processing route. The cube ({001} <100>) and rotated cube ({001} <110>) textures dominate the crystal orientation from the slab until the intermediate annealing stage. After final annealing, there is a texture evolution in both materials; the $\gamma$-fibre component dominates the texture, which is much more intense in modified material supported by components that show good deep drawability, {554} <225>, and good transition from hot to cold rolling, {332} <113>. The modified composition and process material delivers a better re-crystallisation status and, therefore, the best drawability performance.

**Keywords:** forming; texture; fibre; martensite; anisotropy and electron backscatter diffraction (EBSD)





## 1. Introduction

In comparison to austenitic stainless steel, ferritic stainless steel is cheap, price-stable and has good engineering properties. Specifically, EN 1.4016 is used in a large range of applications, including the most common related to formability, stretching and deep drawing [1]. Furthermore, recent studies have aimed to evaluate its applications as a hydrogen container [2], in hospitality and architectural structural elements [3], and in the food and automotive sectors (car interior decoration) [4].

Each rolling and annealing process during the manufacturing of these steels leads to certain preferred crystallographic orientations as a result of their texture, i.e., the sum of the specific crystal orientations in the assemblies of grains within a polycrystalline material [5]. Due to the impact that crystallographic texture has on the anisotropy of material properties, it is necessary to know the mechanisms that are involved in the texture of these materials during its manufacturing route [6–11]. In this sense, the desired $\gamma$-fibre texture associated with Fe-Cr steels occurs during recrystallization and is described by <111> parallel to Normal Direction (ND), thus resulting in good deep drawability and improved strength and toughness. On the other hand, $\alpha$-fibre is a cold rolling texture component that transforms in such a way that <110> orientates parallel to the Rolling Direction (RD), hence reducing formability [12–17].

Measurement of the orientations of polycrystalline grains ($\gamma$ and $\alpha$ phases) after material deformation allows the evaluation of the deformation mechanism. Furthermore, a representation in a three-dimensional space is necessary to describe crystallographic textures. This representation is done by way of ODF sections or diagrams, which are defined by Euler angles $\varphi_1$, $\Phi$, $\varphi_2$ as the consecutive rotations (angles) about the ND, the rotated RD and the rotated ND, respectively, that bring the sample coordinate system (RD, TD, ND) into coincidence for directions relative to the crystal coordinate system ([100], [010] and [001]). Figure 1a is a cross-section of the Euler space, located at $\varphi_2 = 45°$, which shows the most important ideal orientations associated with the plain-strain processing of body-centred cubic (bcc) steels. Figure 1b, in turn, corresponds to an ODF section of the materials analysed and shows the typical positions for the $\alpha$ (RD) fibre and $\gamma$ (ND) fibre textures. The $\alpha$-fibre (low energy nucleation position) is described with {001} <110>, {112} <110> and {111} <110> texture components, whereas the $\gamma$-fibre texture (high energy nucleation position) is described with {111} <110>, and {111} <112> texture components [18]. Other positions evaluated in this study are those related to the presence of solidification structures like {001} <110> rotated cube and {110} <110>, which involve irregular grain growth during heat treatment. Similarly, the orientations of nuclei originating from deformation bands are concentrated around the Goss component, {110} <100> [19,20].

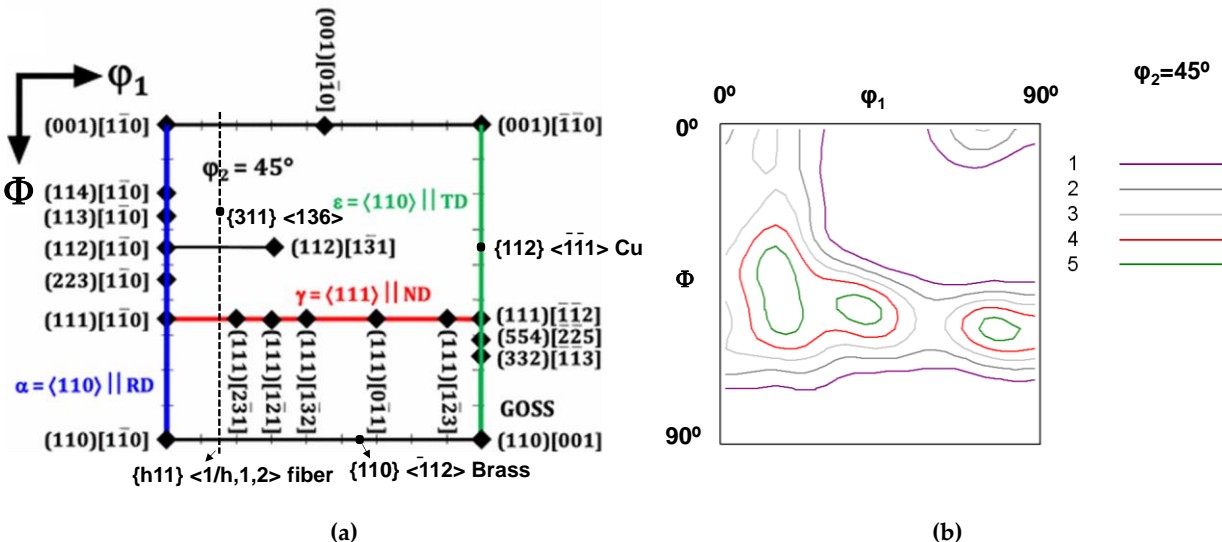

**Figure 1.** (**a**) Plan views of the $\varphi_2 = 45°$ section of Euler space (Bunge notation) applicable to bcc materials subjected to plane strain and (**b**) $\varphi_2 = 45°$ ODF section of the material in this study that shows typical $\gamma$ and $\alpha$ fibres.

Components that occur during the phase transformation of austenite to ferrite in the hot rolling process are {001} <100> Cube, {110} <112> Brass, {332} <113>, {112} <111> Copper and {113} <110>. During subsequent cold rolling, the following components could exist: {113} <110>, {112} <110> and {332} <113>. Among the transformation texture components, the most beneficial from the point of view of achieving good deep drawability is generated after final annealing via the components {111} <110>, {111} <112> and {311} <136> [21–23]. In addition, development of the {554} <225> $\epsilon$ component is ideal for deep drawing [24,25].

Additionally, the anisotropic behaviour of flat products can be evaluated using the Lankford coefficients, with high values of normal plastic anisotropy, $r_{\mathrm{m}}$, greater than one, and planar plastic anisotropy, $\Delta r$, as close to zero as possible, being good indicators of high formability. An increased Lankford value is associated with the development of a <111>||ND $\gamma$-fibre texture during processing of EN 1.4016. On the other hand, the <110>||RD $\alpha$-fibre texture has a low r-value and, consequently, poor drawability during deep drawing of this alloy [26–29].

The heterogeneous texture distribution develops from the initial as-cast structures: equiaxed grains with a random texture will evolve to a homogeneous $\gamma$-fibre, and columnar ones will develop an inhomogeneous texture with predominance of the Goss and $\alpha$-fibre textures [30]. Moreover, texture in steel depends, amongst other variables, on chemical composition, the remains of the primary solidification structure in the final product [31–34], and processing parameters such as finishing temperature during hot rolling, percentage of cold work reduction, and final annealing temperature and time. Consequently, control of the hot rolling process could provide favourable textures, and subsequent cold rolling and annealing can enhance the deep drawing properties of the steel sheet.

Control of the hot rolling process and parameters requires an exhaustive study of EN 1.4016 stainless steel, based on transformation of the ferrite-austenite phase as a function of the rolling temperature. As has been reported for EN 1.4016 steels, the phase transformation depends on the chemical composition of the alloy, and especially on the content of austenite-stabilizing elements such as N, C and Ni, which promote austenite formation at high temperatures. These alloys consist of austenite and ferrite during this rolling process [35,36]. It is expected that austenite will transform into ferrite via diffusion-controlled nucleation and growth. The molar fraction of the transformed austenite influences the microstructure evolution and, therefore, the texture developed [37,38]. Finally, austenite turns into martensite after quenching when it reaches the Ms (Martensite start temperature) during cooling, at which point the martensitic phase starts to nucleate.

In this study, two variations of the same grade of ferritic stainless steel have been analysed. The first type, labelled as 0A, has both the basic EN 1.4016 composition and processing route intended for a broad and general scope of applications. The other one, type 1C, features optimized settings in terms of both chemical composition and manufacturing process to improve the formability properties of EN 1.4016 steels [39].

The technical approach, which has been presented in preliminary studies [40–42], includes characterization by Field Emission Gun–Scanning Electron Microscopy (FEG–SEM) to study texture formation, evolution and microstructural phases obtained during different steps of the processing route, namely casting, intermediate hot rolling, annealing and pickling after hot rolling, and final annealing after cold rolling. Additionally, finishing mill specimens are tested using the Gleeble 1500 thermo-mechanical system as a function of the temperature range for the final rolling step [43].

EBSD post-processing procedures are used to characterize the anisotropy of the material. The ODF is used to analyse the preferred orientation and type of main fibre present in the material. To calculate the content of martensite within the ferrite microstructure, AZtec Reclassify Phase software (4.2 SP1, Oxford Instruments plc, Abingdon (Oxfordshire), UK) [44–48] is used with special features, such as the "band contrast" and "band slope" electron backscatter diffraction patterns (EBSP) quality parameters.

## 2. Materials and Methods

Two EN 1.4016 steel samples with different initial compositions, namely basic (identified as 0A) and modified (1C), were taken from the daily production of Acerinox Europa S.A.U. [49] at different stages of the processing route. In particular, the hot rolling mill of Acerinox Europa S.A.U. was the first in the world to be designed to work with stainless steels, in 1985. At [49] one can obtain more information about this installation. The numbers in Figure 2 indicate the position at which the material was taken for characterization: (1) from slabs during casting, (2) from intermediate hot rolling with physical simulation of the finishing mill, (3) from material after hot rolling with intermediate annealing, (4) after cold rolling and (5) after final annealing. All samples were subjected to thickness reductions between the different production stages: 98% from slab to hot rolling in both materials, and 80% and 87% from hot rolling until final annealing for the basic and modified samples, respectively. In sample 1C, the final annealing treatment processing speed in the carry-in time into the furnace is 20% higher than the annealing applied to sample 0A and this means

that the equivalent time in the furnace, during heat treatment, for sample 1C is higher than for sample 0A.

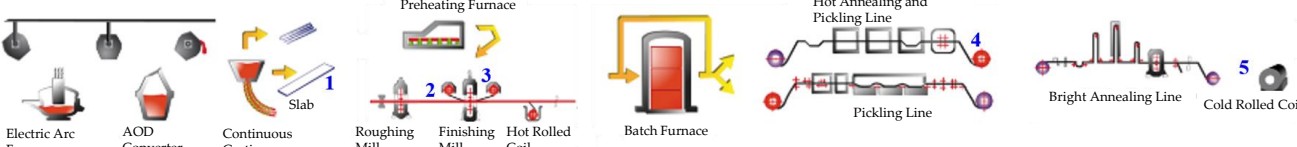

**Figure 2.** Processing route of Acerinox Europa S.A.U. for production lines in study. Numbers in blue indicate the position at which the samples were taken.

The chemical compositions of the samples were analysed using conventional techniques, X-ray fluorescence (XRF), spark optical emission (S-OES) and instruments available in a typical stainless steel factory, using a Panalytical Axios Fast XRF spectrometer (Malvern Panalytical, Malvern, United Kingdom) [50], an OBLF Qsn 750 (OBLF, Witten, Germany) [51] and LECO CS 600 (Leco, Michigan, MI, USA) and TC 600 analysers (Leco, Michigan, MI, USA) [52]. The ThermoCalc software (TCW 4 using database TCFE5) (TC v4, ThermoCalc Software, Stockholm, Sweden) was used to calculate the phase diagrams [53].

The materials selected for this study are summarized in Table 1, together with their global chemical composition and type of annealing applied. The composition of sample 0A is referred to as the "basic" composition, with a high content of austenite-stabilizing elements (C, N, Ni), while sample 1C is a "modified" composition, with a lower content of interstitial elements such as C and N.

**Table 1.** Composition, annealing treatment and metallurgical properties of the as-supplied materials.

| Sample Identification | Composition (wt. %) | Annealing Time |
|:---:|:---:|:---:|
| **0A** | Basic: C: 0.050; N: 0.035; Si: 0.35; Cr: 16.3; S: 0.0009; Mn: 0.41; P: 0.023; Ni: 0.15; Mo: 0.009; Al: 0.003 | Standard |
| **1C** | Modified: C: 0.025; N: 0.025; Si: 0.45; Cr: 16.7; S: 0.0037; Mn: 0.31; P: 0.028; Ni: 0.22; Mo: 0.014; Al: 0.005 | Long |

After the cross-sectional area of the sample taken from the slab, the macrostructure was determined by etching the samples using aqua regia (20 mL $HNO_3$ and 60 mL HCl) by immersion during several min. The microstructure was determined after polishing and etching the samples using Vilella's reagent (5 mL HCl, 1 g picric acid and 100 mL ethanol (95%)) by immersion during 30 s to 75 s, according to standard ASTM E407-07 (2015). Observation was carried out using an Olympus GX71 Light Optical Microscope (LOM) (Olympus, Hamburgo, Germany) [54].

Mechanical tests were carried out for samples with the basic and modified compositions, after hot rolling and final annealing, using a RK100 Roell (ZwickRoell, Ulm, Germany) + Korthaus Universal Testing Machine (ZwickRoell, Ulm, Germany) to study the average Lankford $r$-value and the planar anisotropy. The average anisotropy ($r_m$) and planar anisotropy ($\Delta r$) are calculated using the following equations:

$$r_m = (r_0 + r_{90} + 2r_{45})/4 \tag{1}$$

$$\Delta r = (r_0 + r_{90} - 2r_{45})/2 \tag{2}$$

Thermo-mechanical physical simulation of the hot rolling finishing mill (FM) process was carried out in a Gleeble 1500 D system (DSI, New York, NY, USA), applying mechanical and thermal cycles to cylindrical specimens. Three high temperature tensile tests were

executed for each material, and each test was performed at the temperature of the first, intermediate and last rolling pass of the FM train, respectively.

EBSD characterization was performed with a Zeiss Ultra 55 FEG-SEM (Zeiss, Oberkochen, Germany) [55], using polished longitudinal sections of the samples. The sample sections were prepared as per conventional metallographic procedures, with a final polish using colloidal silica suspension OPS (Oxide Polishing Suspension) [56]. The FEG-SEM used was equipped with a CHANNEL 5 EBSD system (Oxford Instruments plc, Abingdon (Oxfordshire), UK), from Oxford Instruments [57]. EBSD maps were acquired at 20 kV, at a working distance of 16 mm and a 0.5 μm step size, and using the post-processing maps software Tango (4.2 SP1, Oxford Instruments plc, Abingdon (Oxfordshire), UK) and Salsa (4.2 SP1, Oxford Instruments plc, Abingdon (Oxfordshire), UK). For the grain boundary component, Tango produces a correlated misorientation angle distribution that makes it possible to obtain the content of grains, subgrains and twin boundaries. The Tango recrystallized fraction component detects deformed and recrystallized grains, and therefore the content of these fractions in the analysed area. Salsa is the tool used to generate an ODF, an approach to texture representation.

In addition, the Aztec Reclassify Phase software (4.2 SP1, Oxford Instruments plc, Abingdon (Oxfordshire), UK), which takes into consideration the austenite–ferrite transformation phenomenon, allows the content of martensite within the ferrite matrix to be calculated from the EBSD analysis, as mentioned in the previous section. Martensite and ferrite cannot be discriminated by EBSD. To overcome this limitation, this software uses the density of crystalline defects to identify the martensite areas as follows in Table 2: the tetragonal elongation, which is characteristic of the martensite microstructure, between the a, b and c lattice parameters of both microstructures is greater than 3%, hence there is a real chance to separate martensite from Fe bcc by "Refined Accuracy", as explained in references [45,58]; in this case it would be higher than 3% (3% = 2.9313 c). The specific algorithm to solve the pattern is described in the references [45,47,48]. The use of martensite structural data makes it possible to recognize and index martensite in a range greater than 3% of tetragonal distortion [59]. The martensite content calculated is a measure of occupied area in the microstructure analysed. The presence of martensite was then checked by light optical microscopy [60].

**Table 2.** Fe bcc and Martensite crystallographic parameters; Inorganic Crystal Microstructure Database (ICSD) pattern.

| Fe bcc | | Martensite | |
|---|---|---|---|
| Crystal System: | Cubic | Crystal System | Tetragonal |
| Space Group: | Im-3m | Space Group: | I4/mmm |
| Space Group number: | 29 | Space Group number: | 139 |
| a (nm): | 0.2871 | a (nm): | 0.2846 |
| b (nm): | 0.2871 | b (nm): | 0.2846 |
| c: (nm): | 0.2871 | c: (nm): | 0.3053 |
| Alpha (°): | 90.0000 | Alpha (°): | 90.0000 |
| Beta (°): | 90.0000 | Beta (°): | 90.0000 |
| Gamma (°): | 90.0000 | Gamma (°): | 90.0000 |
| Calculated density (g/cm$^3$): | 7.8300 | Calculated density (g/cm$^3$): | 7.0900 |
| Volume of cell ($10^6$ pm$^3$): | 23.6600 | Volume of cell ($10^6$ pm$^3$): | 24.7300 |
| Z: | 2.00 | Z: | 1.00 |
| RIR: | 11.07 | RIR: | 7.36 |

Taking into account the experimental procedure, Table 3 summarises the identification of the samples used in this study, along with their corresponding position in the manufacturing process. Also, Figure 3 shows the processing temperature of both materials in the following positions of the route: Hot Rolling (HR), Roughing Mill outside (RMout), Finishing Mill Initial (FMin) and Finishing Mill outside (FMout); Intermediate Annealing (IA); and Final Annealing (FA).

**Table 3.** List of samples studied and positions along the stainless steel processing route. Numbers are positions marked in Figure 2.

| Sample Id. | Position |
|---|---|
| 0A-S | Slab (1) |
| 1C-S | Slab (1) |
| 0A-HR | Hot Rolling (2): Roughing Mill out |
| 1C-HR | Hot Rolling (2): Roughing Mill out |
| 0A-FM | Hot Rolling: Finishing Mill (3) |
| 1C-FM | Hot Rolling: Finishing Mill (3) |
| 0A-IA | Intermediate Annealing (4) |
| 1C-IA | Intermediate Annealing (4) |
| 0A-FA | Final Annealing (5) |
| 1C-FA | Final Annealing (5) |

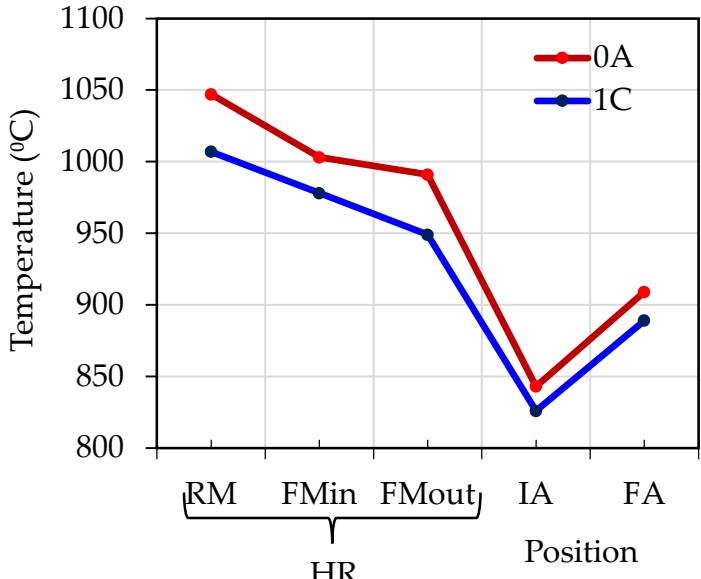

**Figure 3.** Processing temperature for production lines in study for both materials. Hot Rolling (HR); Roughing Mill (RM); Finising Mill initial (FMin); Finishing Mill outside (FMout); Intermediate Annealing (IA); Final Annealing (FA).

## 3. Results and Discussion

### 3.1. Phase Diagram

The phase diagrams of the chemical compositions studied were calculated using ThermoCalc software. Figure 4 shows the phase diagram of the basic (0A) and modified (1C) samples, which are characterized by a dual-phase ferrite–austenite structure gamma loop and precipitation of chromium carbide type $M_{23}C_6$.

The phase diagrams calculated using ThermoCalc show important differences in the two materials in terms of transformation points and amount of high temperature austenite: Ac1 and Ac3 are 855 °C and 1256 °C, respectively, for the 0A material, and 857 °C and 1180 °C, respectively, for the 1C material, while the maximum austenite contents are 53.3% (at 919 °C) for 0A and 32.8% (at 943 °C) for 1C. As this austenite is not stable, it transforms into martensite when cooling to room temperature. The higher the content of austenite-stabilizing elements, the greater the transformation.

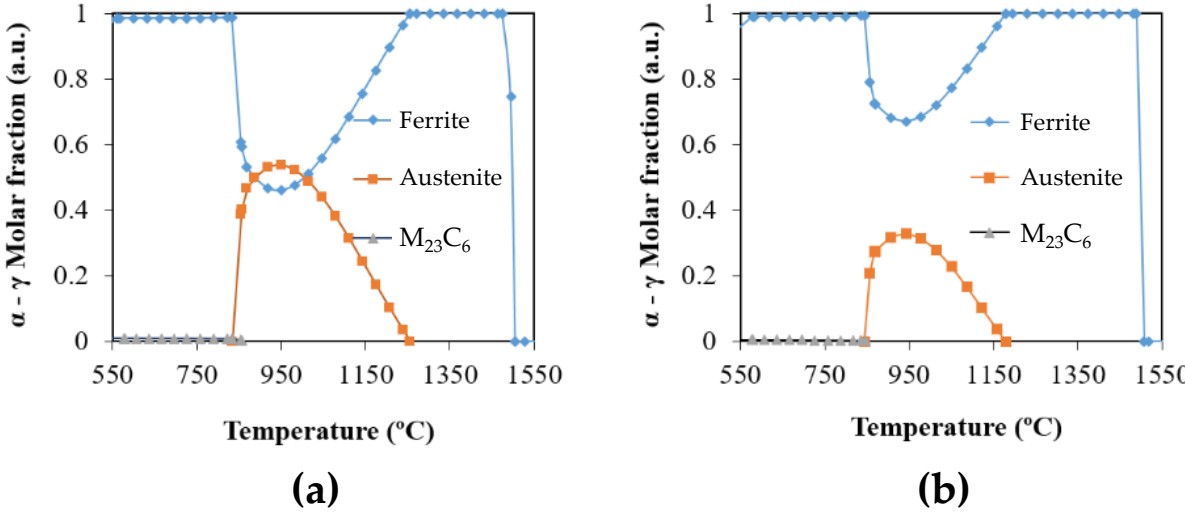

**Figure 4.** Phase diagrams for the material systems in sample 0A (**a**) and sample 1C (**b**).

### 3.2. Microstructure and Macrostructure at Casting Stage

Figure 5 shows Light Optical Microscopy (LOM) images of the microstructure in samples from slabs for basic 0A-S (Figure 5a) and modified 1C-S (Figure 5b) material. The optical micrographs of both slab microstructures show ferritic grains with martensite needles at grain boundaries and within the matrix. Moreover, subgrains can be seen in the magnified micrographs (Figure 5c) for the basic composition material (0A-S) but not the modified one (1C-S). Additionally, the distribution of carbides, which are dispersed within the grains, is noticeable (Figure 5d). As such, the microstructure of material 1C-S shows a better recrystallization state.

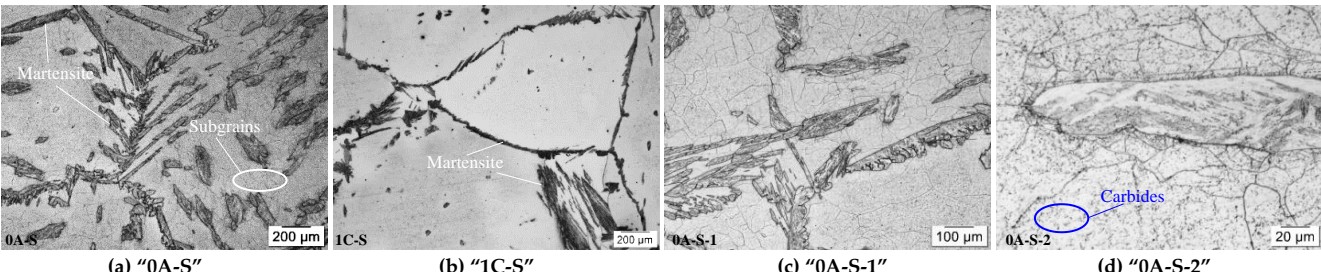

**Figure 5.** Light Optical Microscopy (LOM) images showing the microstructure of slabs 0A and 1C. (**a**) "0A-S"; (**b**) "1C-S"; (**c**) "0A-S-1"; (**d**) "0A-S-2".

Figure 6 shows a scheme of the samples taken from the slab for macrostructural characterization (in blue) and for EBSD analysis parallel to RD (casting direction; in red), along with a magnified view of the macrostructure for each sample (0A-S and 1C-S), in order to study the solidification mechanism of both ferritic stainless steels to assess their cooling mechanism. To that end, the cooling rate from both surfaces to the core of each slab was observed.

Material 0A shows the presence of four zones: (from bottom to top) a columnar zone containing large crystals grown along the direction of solidification, followed by a mixed equiaxed and columnar crystal zone; then, an equiaxed crystal zone, and finally, a columnar zone. In contrast, material 1C shows the typical macrostructure: columnar grains grown from surfaces through the core of the slab and an equiaxed crystal zone at the centre. As such, the macrostructure characterization indicates that the solidification mechanism is heterogeneous in the material with basic composition, but homogeneous in the modified composition. The undesired columnar grains that form among equiaxed grains in 0A result in a less uniform microstructure. On the other hand, material 1C presents a larger area of

columnar grains, which are known to develop a heterogeneous distribution of texture {100} parallel to ND during thermo-mechanical processes, while a random texture of equiaxed solidification microstructures leads to the homogeneous γ-fibre during the rolling and annealing process [30].

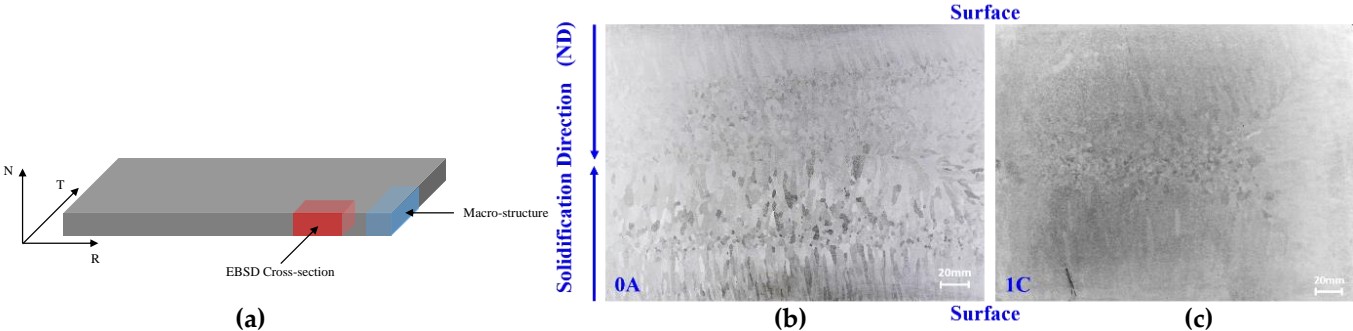

**Figure 6.** Scheme of the samples taken from the slab (**a**) showing rolling (R), normal (N) and transversal (T) axes, and the macrostructure in the basic 0A-S (**b**) and modified materials 1C-S (**c**). The size of each slab is $260 \times 190$ cm$^2$.

### 3.3. Simulation of a Finishing Mill

This simulation allows the ductility of the materials to be studied under the temperature conditions of a hot rolling mill. Under normal circumstances of hot rolling, the thickness reduction and temperatures in this process differ from one sample to the other. Therefore, in order to determine the actual plastic behaviour of 0A and 1C steels at rolling temperature, high temperature tensile tests were carried out on three specimens of each stainless steel in a Gleeble system (point 3 in Figure 2). These tests were executed at the different rolling pass temperatures corresponding to the initial (I), medium (M) and final (F) rolling of the finishing mill process. The plastic behaviour at high temperature was studied in terms of ductility, which depends on temperature and steel microstructure. The latter is associated with the degree of ferrite–austenite transformation that takes place in 0A and 1C steels as a function of rolling temperature. Therefore, this study is able to establish the influence of rolling temperature on the ferrite–austenite transformation, according to the ductility of each specimen during simulation of the hot rolling process.

Simulation parameters: I, M and F temperatures for 0A material, 920 °C, 910 °C and 885 °C respectively, and for 1C material, 885 °C, 870 °C and 820 °C respectively; the temperature program is applied for 2 min until the target temperature for each material is reached; the strain distance is 10 mm; the strain strength applied to the specimen is 50 Hz; and the strain time is 4.76 s until failure. After testing, the ductility (deformability) is determined from the percentage reduction of the transversal area of strain direction or diameter (%AR, %DR); the smaller the final area or diameter of the transversal section, the higher the ductility. Further results (not shown here) might be obtained from this test regarding maximum resistance, elongation and microstructural evolution of longitudinal deformation.

Figure 7 shows the ductility of both materials vs. rolling temperature. The ductility of 0A-FM decreases slightly and then increases rapidly with decreasing temperature, whereas the ductility of 1C-FM decreases strongly at medium rolling temperature and continues to decrease with temperature. The temperature shift displayed in both materials in Figure 7 is due to the fact that 0A and 1C ferritic stainless steels are processed at different finishing mill temperatures industrially.

In order to compare the high temperature ductility between 0A and 1C steels at the same temperature, additional tension tests were carried out while maintaining the temperature at 900 °C. The results show that 0A-FM exhibits better plastic behaviour, with a reduction of the final area of the transversal section of 98% and 99%, than 1C-FM, with a 95% reduction. According to Figure 4, the microstructure of the 0A and 1C steels at 900 °C

is based on a ferrite–austenite duplex state, with the amount of austenite (more ductile phase) being greater in the basic 0A steel. Generally, high temperatures lead to a change in the microstructure according to the alloy chemical composition. In 0A steel, the lower Cr composition (16%) and higher C content (0.044%) cause the steel microstructure to be highly duplex, which results in higher ductility during the hot rolling process compared to 1C steel.

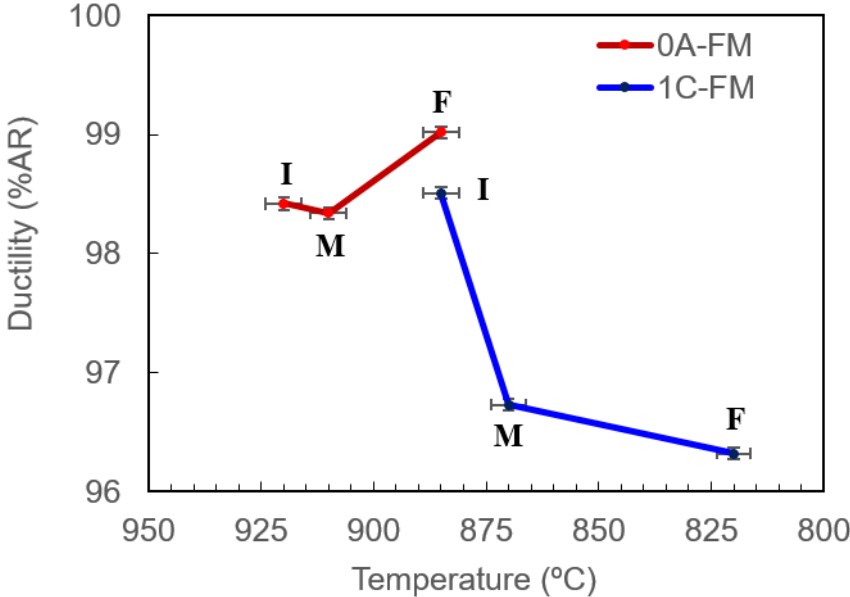

**Figure 7.** Ductility of 0A and 1C as a function of rolling temperature. Tensile testing temperatures are set in order to reproduce industrial conditions (Initial (I), Medium (M), Final (F)). Temperature range based on the manufacturing direction.

### 3.4. Anisotropy and Strain Hardening

The results of anisotropy and strain hardening of both, hot-rolled with intermediate (IA) and final annealed (FA) materials, are shown in Table 4.

**Table 4.** Anisotropy and strain hardening results for 0A and 1C with Intermediate Annealing (IA) and Final Annealing (FA).

| Sample Id. | *r*-Values [1] | | | $r_m$ [5] | $\Delta r$ [6] | $n$ [7] |
|---|---|---|---|---|---|---|
| | RD [2] | ND [3] | TD [4] | | | |
| 0A-IA | 0.36 | 0.63 | 0.48 | 0.52 | −0.21 | 0.18 |
| 1C-IA | 0.34 | 0.69 | 0.54 | 0.57 | −0.25 | 0.15 |
| 0A-FA | 0.86 | 0.70 | 1.18 | 0.86 | 0.32 | 0.21 |
| 1C-FA | 0.80 | 1.05 | 1.21 | 1.17 | −0.09 | 0.21 |

[1]—Lankford coefficients or strain ratios; [2]—Rolling Direction; [3]—Normal Direction; [4]—Transversal Direction; [5]—Average anisotropy; [6]—Planar anisotropy; [7]—Tensile strain-hardening exponent.

The normal anisotropy and strain hardening values in IA materials are significantly lower than those for the FA materials, which is consistent with the corresponding microstructures: deformed and recrystallized. On the other hand, both normal and planar anisotropy values are clearly more favourable in the 1C material, which means a better performance in deep-drawing operations. Consequently, the basic composition sheet exhibits a more pronounced anisotropy and lower deep drawability than the modified composition.

### 3.5. EBSD Analysis along the Processing Route

Characterization of the microstructure by analysing the content of grain boundaries, recrystallized and martensite areas, and texture evolution by interpreting the ODF, was performed for basic (0A) and modified (1C) materials at the steps in the processing route marked in Figure 2. The samples are identified as shown in Table 3.

Table 5 summarizes the orientation density results of the studied samples, quantifying the colour scale of each ODF, and showing how strongly a particular orientation appears. These values were measured with respect to the ideal location of the texture component of fcc and bcc metals [8], which is obtained from the ODF shown in Figure 1, and analysed in the ODF of each sample obtained from the processing route: slab, hot rolling, intermediate annealing and final annealing. In steel textures, it should be clarified that the maximum intensities along the fibre are not in the exact position of the ideal fibre and that there are usually deviations of up to 15° [18].

**Table 5.** Orientation density results for textures evaluated in the $\varphi_2 = 45°$ section of Euler space.

| Texture Component | Euler Angles $(\varphi_1, \Phi), \varphi_2 = 45°$ | 0A-S | 1A-S | 0A-HR | 1C-HR | 0A-IA | 1C-IA | 0A-FA | 1C-FA |
|---|---|---|---|---|---|---|---|---|---|
| {001} <110>, Rotated Cube | (0°, 0°) | 0.43 | 0 | 0.93 | 4.78 | 7.51 | 19.50 | 1.00 | 1.27 |
| {001} <110>, Rotated Cube | (90°, 0°) | 0.36 | 0 | 0.87 | 4.85 | 7.73 | 19.60 | 1.00 | 1.25 |
| {110} <110> | (0°, 90°) | 0 | 1.62 | 0.41 | 0 | 0 | 0.02 | 0.07 | 0 |
| {001} <100>, Cube | (45°, 0°) | 1.97 | 0 | 0 | 0.40 | 3.05 | 0.28 | 0.79 | 0 |
| {110} <100> Goss | (90°, 90°) | 0 | 0.04 | 1.19 | 2.04 | 6.42 | 0.61 | 1.60 | 0 |
| {110} <112>, Brass | (55°, 90°) | 0.59 | 0 | 0.51 | 0.30 | 0 | 0 | 0.20 | 0.35 |
| {332} <113> | (90°, 65°) | 2.59 | 0 | 1.20 | 0.55 | 0.08 | 0.26 | 2.06 | 7.70 |
| {112} <111>, Copper | (90°, 35°) | 1.63 | 0.67 | 0.57 | 0.23 | 0.06 | 0 | 0.32 | 0.39 |
| {113} <110> | (0°, 25°) | 0.05 | 2.08 | 0.89 | 0.58 | 3.17 | 7.68 | 1.21 | 1.35 |
| {112} <110> | (0°, 35°) | 0 | 0 | 0.65 | 0.62 | 1.33 | 1.74 | 0.61 | 0.54 |
| {554} <225> | (90°, 60°) | 0.13 | 0 | 1.36 | 0.35 | 0 | 0.39 | 2.62 | 14.20 |
| {111} <110> | (0°, 55°) | 0.52 | 0 | 0.67 | 0.24 | 1.08 | 0 | 2.31 | 2.85 |
| {111} <112> | (30°, 55°) | 0.34 | 0.13 | 0.89 | 0.30 | 0 | 0.49 | 1.76 | 12.60 |
| {111} <110> | (60°, 55°) | 0.38 | 0 | 0.76 | 0.25 | 1.29 | 0 | 2.62 | 2.86 |
| {111} <112> | (90°, 55°) | 0.28 | 0.08 | 0.97 | 0.29 | 0 | 0.48 | 1.93 | 13.40 |
| {311} <136> | (15°, 25°) | 1.06 | 0 | 1.24 | 1.27 | 1.0 | 5.68 | 2.53 | 2.50 |

#### 3.5.1. Casting

A specimen for EBSD analysis was taken from each 0A-S and 1C-S slab in the direction parallel to the continuous casting direction. According to Figure 8, the specimens correspond to the elongated columnar microstructure, which is the most abundant area for each slab and comprises non-random oriented grains [30]. The acquisition parameters used to prepare the EBSD map depend on the size of each sample and, for the 0A-S material, this map presents a grid of 5988 × 43,137 pixels and a step size of 5.0 μm, thus giving an area of 29,900 × 15,700 μm². A grid of 9231 × 3405 pixels and a step size of 3.0 μm was used in 1C-S to give a rastered area of 27,700 × 10,200 μm².

The characterization results presented in Figure 8 reveal the presence of a phase in the ferrite matrix identified as retained martensite (red areas in images overlapped in the band contrast map, which shows the microstructure of each material in grey), located in inter and intragranular positions, in both materials. The martensite content in the 0A-S material is larger than that in 1C-S, and mainly presents a needle-shaped structure. It can also be seen from Figure 8 that the ferrite matrix grain size in 1C-S is larger than that in 0A-S. The Reclassify phase software was used to calculate the content of this martensite, which is 15% in 0A-S and 6.6% in the 1C-S sample.

Figure 9 resumes the main characteristics of the slab microstructures, as studied by EBSD. Figure 9a quantifies the subgrain content (shown in blue) obtained from misorientation angles, and indicates that 85% of the grain boundaries analysed correspond to a subgrain, with this value being similar in both materials. Quantification of Σ3 (shown in

yellow), which is the well-known twin special boundary in face-centered cubic materials, a specific coincidence of the crystal lattices (Coincidence Site Lattice (CSL)) across the boundary, shows that 63% of the total CSL corresponds to $\Sigma 3$ in the 0A-S material while in the 1C-S one it is 43%. The deformed fraction of non-recrystallised grains, which occurs when the average angle in a grain exceeds the minimum angle to define a subgrain (10° in this investigation), is shown in green. This content is higher in 0A-S (25%) than 1C-S (5%). Finally, the martensite content (shown in red), which is a measure of occupied area in the ferrite matrix, is higher in 0A-1S (15%) than in 1C-1S (7%). Figure 9 also includes the ODF sections obtained for the slab samples.

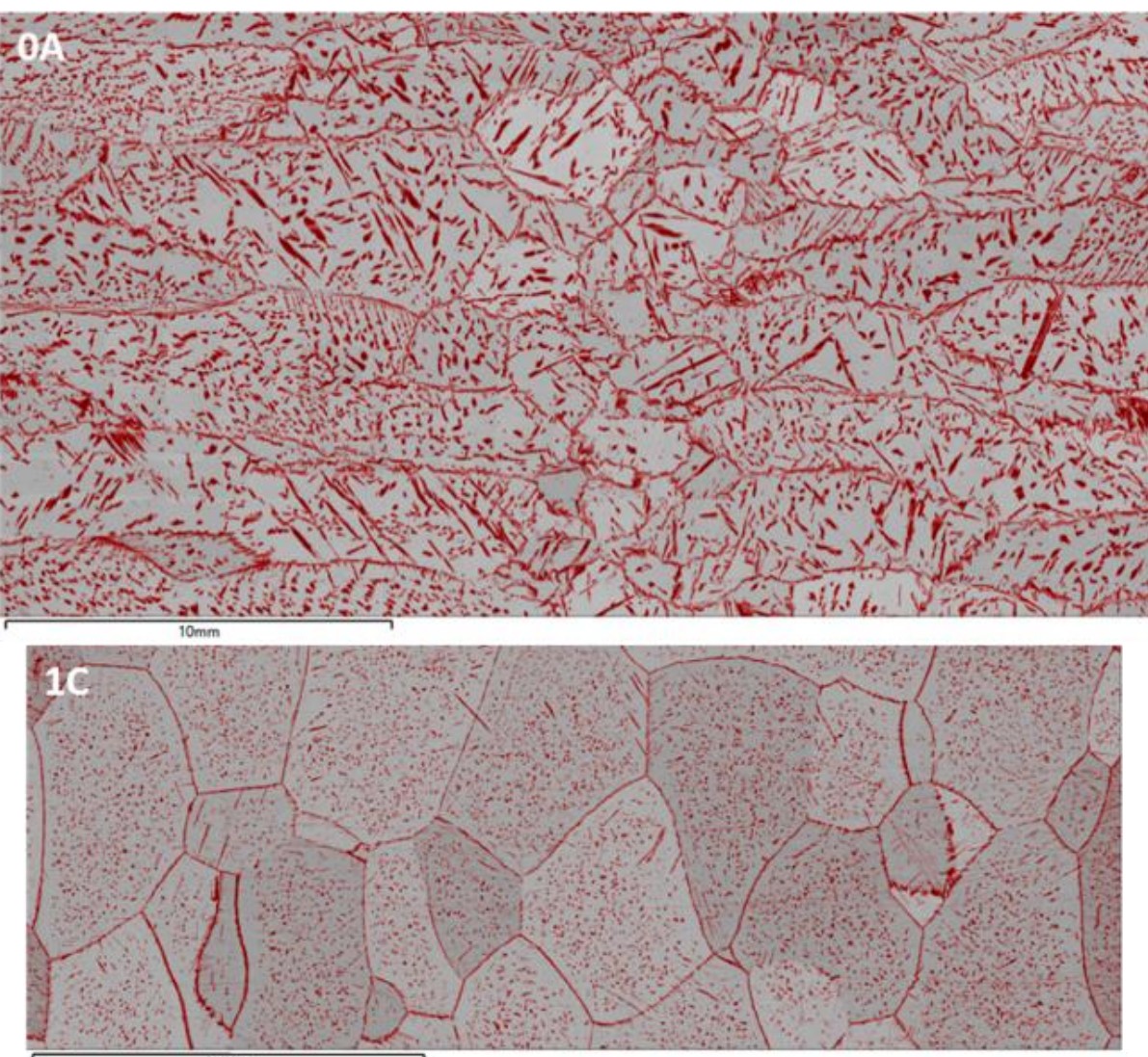

**Figure 8.** Band Contrast (BC) maps for 0A-S and 1C-S. Retained martensite is highlighted in red.

The content of subgrain boundaries is high and similar in both materials, although subgrains were not observed by LOM in 1C-S (Figure 5). Associated with these boundaries, the microstructure of the material can be deformed or sub-structured. The subgrains are located in deformed areas such as martensite, which corresponds to the previous austenite transformed, and also contain the special $\Sigma 3$ joints. The 0A-S material has a higher $\Sigma 3$ content, deformed area and martensite content than 1C-S. As such, subgrains belong to deformed areas, although to a lesser degree than in 1C-S; therefore, the excess of subgrains must be in the ferrite matrix of the 1C-S material. This could produce a dense subgrain

network with higher dislocation movements during uniaxial deformation because of lower precipitation (lower content of interstitial elements, such as C and N) than 0A-S.

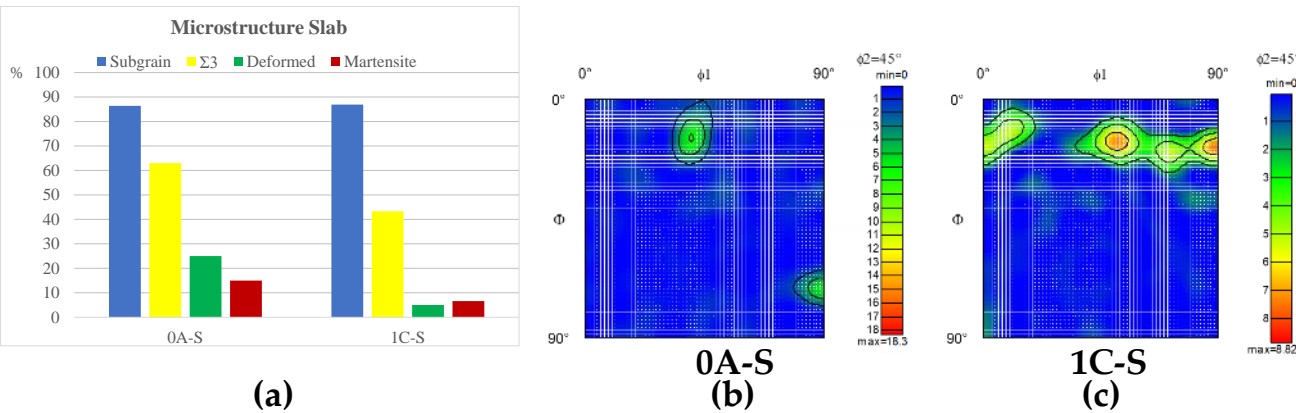

**Figure 9.** Internal structure characteristics of slabs: subgrain, Σ3, deformed fraction and martensite content (**a**); ODF sections for 0A-S (**b**) and 1C-S (**c**).

Figure 9b,c show very weak textures for both materials. Using these ODF in combination with Table 5, and considering deviations of up to 15° in the ODF section, it can be seen that the high orientation density found in 0A-S is related to {001} <100> cube from the initial as-cast columnar microstructures and components such as {332} <113> and {112} <111> copper (twinning system) associated with austenite or the transformed martensite [27]. In 1C-S, the high density corresponds to the rotated cube {001} <110> resulting from the austenite in the ferrite. This suggests that most of the austenite was transformed to martensite by deformation accumulation in the austenite prior to transformation. Furthermore, the texture {113} <110> component, which is related to the presence of martensite, should also be noted.

### 3.5.2. Hot Rolling

During the hot rolling process, the casting microstructure is transformed into a more homogeneous microstructure and the mechanical properties are improved due to recrystallization phenomena that take place during deformation of the steel at high temperatures. In order to determine the microstructure during hot rolling, a sample parallel to the rolling direction was extracted from transfer bars of each steel after the roughing mill and before the finishing mill (position 2 in Figure 2).

Figure 10 shows band contrast maps, obtained from the EBSD measurements, corresponding to 0A-HR and 1C-HR. A grid of 524 × 390 pixels and a step size of 2.1 μm were the acquisition parameters in both hot rolling samples, thus obtaining a rastered area of 1100 × 820 μm$^2$. Grain, subgrain and CSL boundaries are shown in the maps as overlays. Areas in red represent the martensite fraction, in which special Σ3 joints (yellow) are concentrated. 0A-HR has a higher volume fraction of martensite than 1C-HR. This is due to the higher austenite volume fraction in 0A-S with respect to 1C-S. In addition, these areas involve a high content of subgrains and deformed structure.

The microstructures consist of bands of martensite and ferrite, which form a pancake structure and elongate parallel to the rolling direction, mainly in 0A-HR. Figure 11a summarises the Aztec Reclassify Phase software-quantified analysis for the hot-rolled materials. This graph indicates that the content of subgrain boundaries is similar in both materials. However, 0A-HR has a higher Σ3 content than 1C-HR, which is related to the higher martensite and deformed area content with respect to the latter. As is also the case for slab samples, it is expected to find a higher density of subgrains in the ferrite matrix of the 1C-HR material with high dislocation movement due to the lower precipitation in this type of steel (lower content of interstitial elements such as C and N).

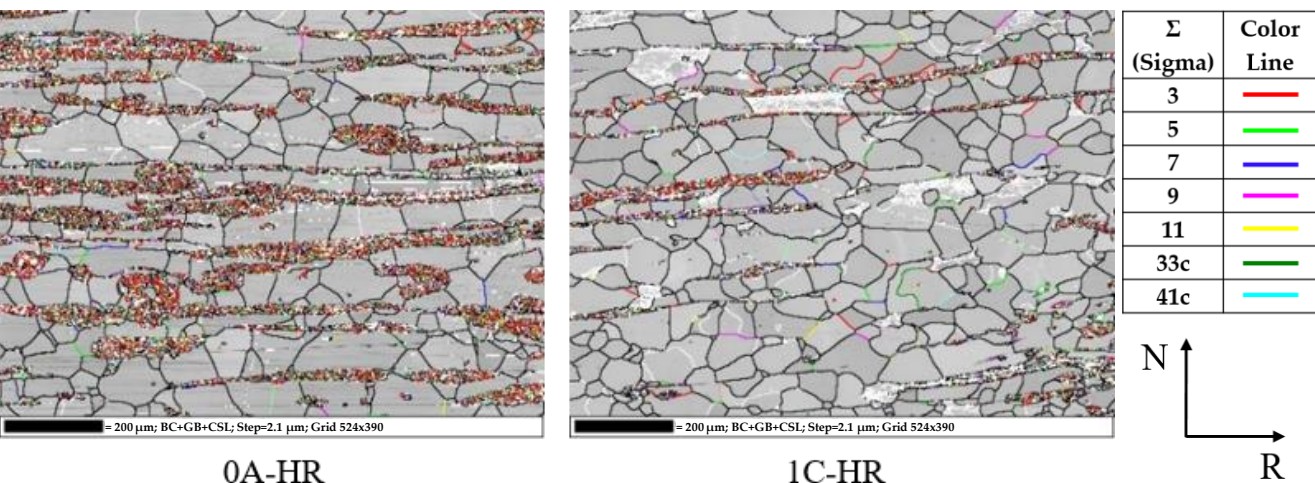

**Figure 10.** BC + CSL + GB map of basic and modified material during hot rolling, in the Normal-Rolling Direction (ND-RD) cross-section. Grid 524 × 390 pixels; step size 2.1 μm; rastered area 1100 × 820 μm$^2$. Also is showed a legend that illustrate the colour lines in the maps, indicating for each colour its corresponding sigma.

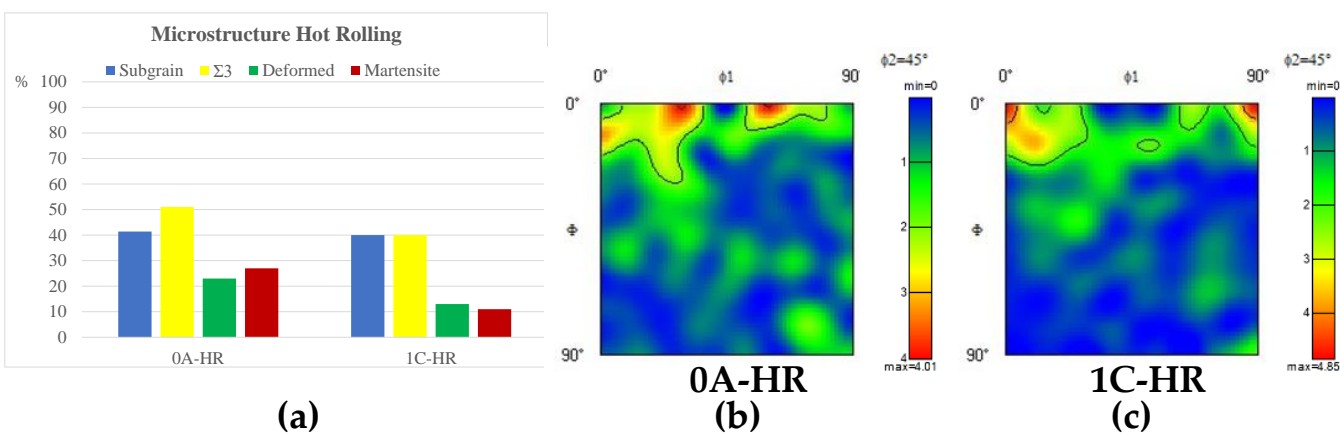

**Figure 11.** Internal structural characteristics of hot rolled material (**a**). ODF 0A-HR (Hot Rolling) (**b**) and 1C-HR (**c**).

With respect to the texture, it can be seen from Figure 11b,c and Table 5 that the initial solidification texture {001} <100> was retained in 0A-HR material and the {001} <110> rotated cube in material 1C-HR. In addition, the presence of {001} <100> in 1C-HR and {001} <110> rotated cube in 0A-HR is observed. Deformation texture bands such as Goss texture and <011> parallel to ND are present in both materials. Moreover, a weak density band associated with the γ-fibre texture <111> parallel to ND is observed in both materials, with higher orientation values for 0A-HR.

### 3.5.3. Intermediate Annealing, after Hot Rolling

During the finishing stage of the hot rolling process, the thickness of each material is further reduced, with a total reduction (with respect to slabs) of 98.0% for 0A-IA and 98.3% for 1C-IA. Finally, these materials are annealed during the same exposure time to regenerate the hot-rolled deformed structure by recrystallisation. Moreover, this study considered manufacturing specifications regarding the finishing mill process. Thus, no load was applied in the last hot-rolling pass of basic material during the finishing mill process, whereas the same process was performed with load for the modified material. This suggests that the structure of the 0A-IA coil had a longer recovery and recrystallisation time than the 1C-IA material.

Figure 12 displays band contrast maps corresponding to 0A-IA and 1C-IA after intermediate annealing and GB and CSL, respectively, as mentioned in Figure 10. In a sample extracted from this step (position 4 in Figure 2), an area of 522 × 393 pixels with a step size of 1.4 µm was analysed, to give a rastered area of 731 × 550 µm$^2$. EBSD analysis does not show the presence of retained martensite in the ferritic matrix in any of the materials after hot rolling and annealing. This analysis showed that the stored energy from the hot-rolling process was used in the austenite–ferrite transformation. On the other hand, the grain size for 1C-IA is significantly higher than for 0A-IA. The presence of biphasic martensite–ferrite in the 0A-IA material during hot rolling could help to maintain the grain size and prevent it from growing excessively, as occurs in the 1C-IA material.

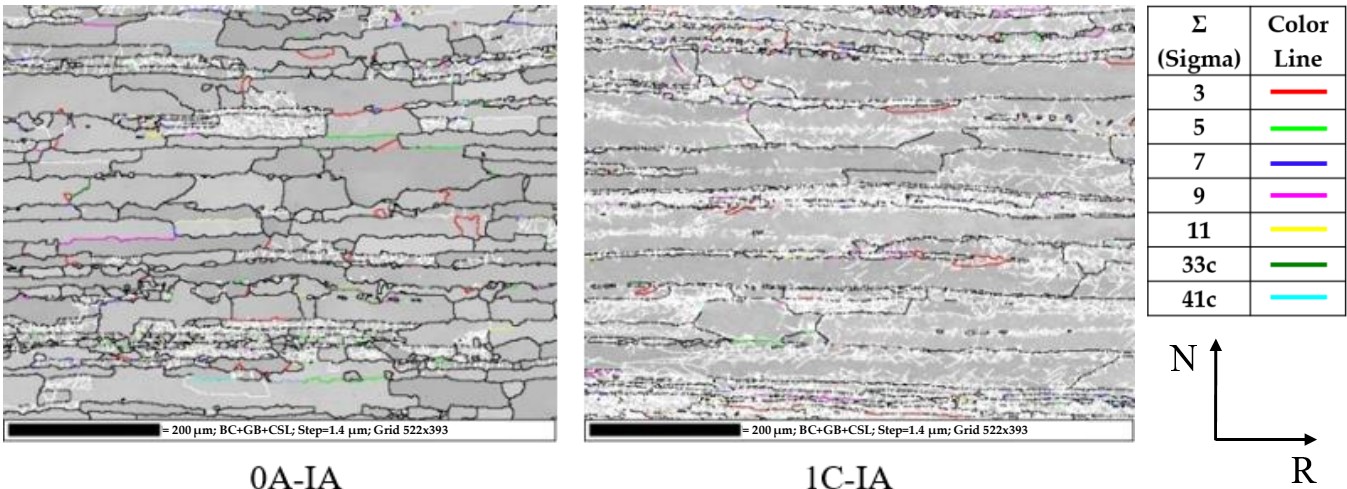

**Figure 12.** BC + CSL + GB map of basic and modified material with intermediate annealing, in the ND-RD cross-section. Grid 522 × 393 pixels; step size 1.4 µm; rastered area 731 × 550 µm$^2$. Also shown is a legend that illustrates the colour lines in the maps, indicating for each colour its corresponding sigma.

The microstructure of 1C-IA showed a recovered state, which contrasts with the partially or fully recrystallised state of the 0A-IA sample. Figure 13a shows that the subgrain fraction of 1C-IA is higher than in 0A-IA, which indicates that the structure of 1C-IA after hot rolling and before annealing is based on a recovered state. Moreover, a poor recrystallisation is observed in the modified material, with 1C-IA having a higher Σ3 and deformed area content than 0A-IA, probably due to the different manufacturing of both materials during the last hot-rolling pass. In the basic material, with no load, the high temperature applied to the steel is used to recover its structure, transforming the austenite generated during the hot rolling process and starting the recrystallisation phenomenon. In contrast, for the modified material, with load, the steel had a shorter time between the last hot rolling pass and the start of the intermediate annealing process, thus resulting in a more deformed structure with respect to the basic material.

The ODFs indicate that α-fibre prevails in both materials, especially in 1C-IA (Figure 13c and Table 5). The cube and Goss deformation texture bands and shear banding [8] present in 0A-IA (Figure 13b) correspond to bcc orientations derived from fcc rolling and recrystallisation components acquired during the martensite transformation. A further component seen in 1C-IA is {311} <136>, which encourages a good transition from hot to cold rolling after annealing, thus producing locally potent nucleation sites for recrystallisation.

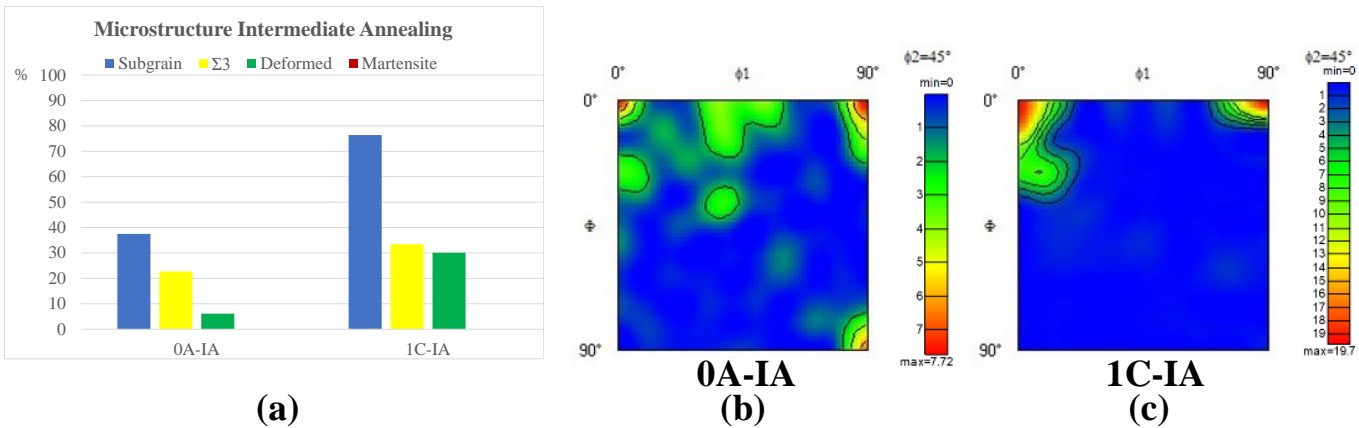

**Figure 13.** Internal properties of material during intermediate annealing (**a**) ODF sections for 0A-IA (Intermediate Annealing) (**b**) and 1C-IA (**c**).

### 3.5.4. After Cold Rolling and Final Annealing

After intermediate annealing, both steels were cold-rolled in order to reach the final thickness. According to industrial practice, 0A-FA experienced a reduction of 80%, whereas 1C-FA reached a 90% thickness reduction. In consequence, the stored energy of cold work was higher in 1C-FA than 0A-FA. In agreement with previous research, the stored energy is the driving force of both recovery and recrystallisation during annealing (position 5 in Figure 2).

This is clearly identified by EBSD, which showed a completely recovered and recrystallised microstructure in 1C-FA (Figure 14). The acquisition parameters are the same in both materials (grid of 548 × 408 pixels and a step size of 0.5 μm), with a rastered area 274 × 204 μm². The band contrast map for 0A-FA, Figure 14 or Figure 15a, show the presence of retained martensite in the ferritic matrix due to the final annealing of this material. Moreover, the gamma loop seen in Figure 4 shows that austenite is not stable and could transform to martensite.

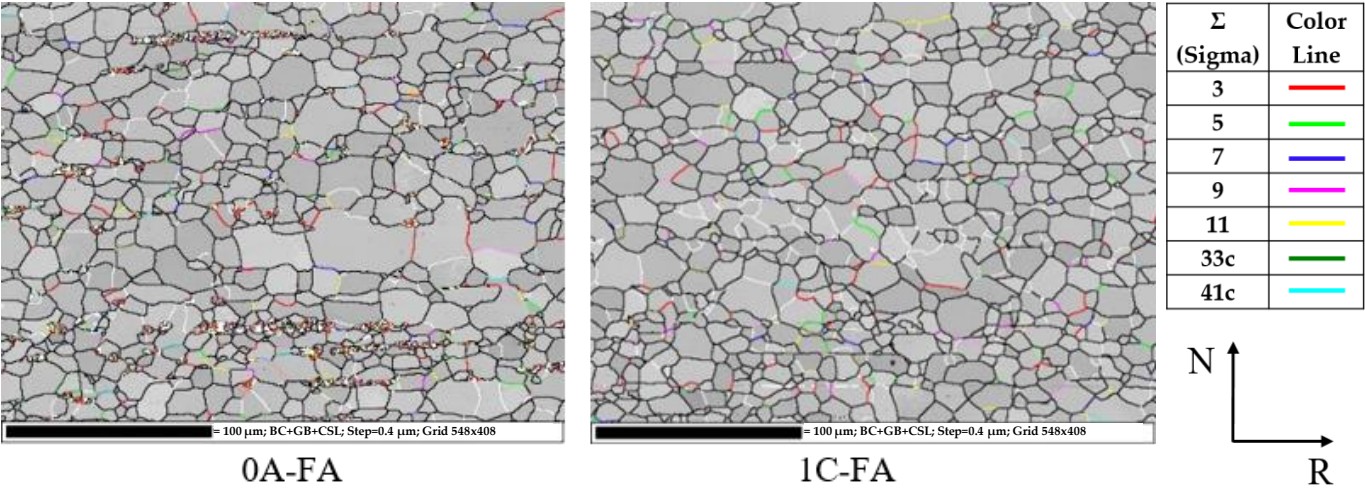

**Figure 14.** BC + CSL + GB map of basic and modified material with final annealing, in the ND-RD cross-section. Grid 548 × 408 pixels; step size 0.5 μm; rastered area 274 × 204 μm². Also shown is a legend that illustrates the colour lines in the maps, indicating for each colour its corresponding sigma.

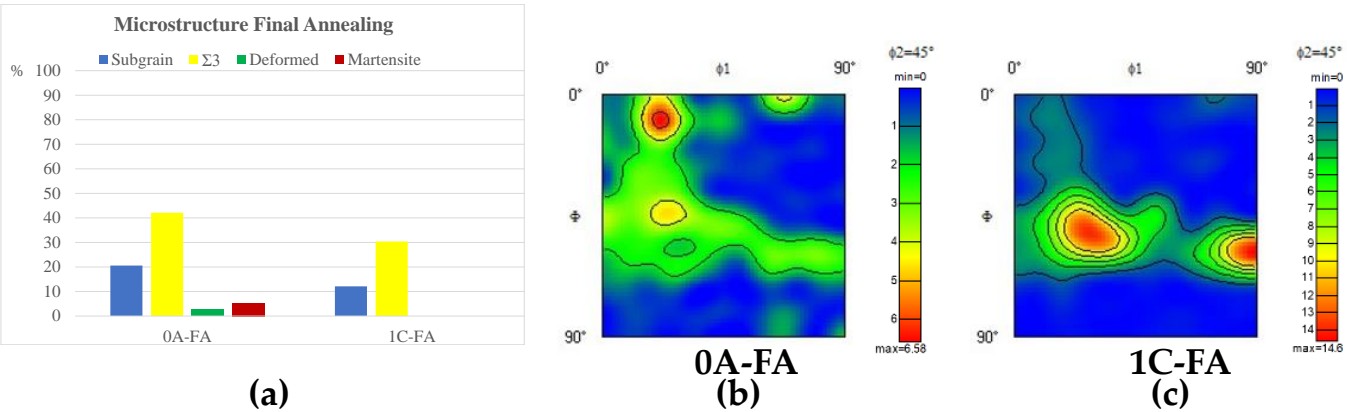

**Figure 15.** Internal properties in final annealing material (**a**) ODF sections for 0A-IA (**b**) and 1C-IA (**c**).

Since the plastic deformation was higher in 1C-FA, this material is expected to achieve a fully recrystallised state after annealing (Figure 15a). Moreover, 0A-FA has a higher subgrain and Σ3 content, which are located in the martensitic deformed areas, than 0A-HR.

A texture transformation occurs in ferritic steels after cold rolling and final annealing. Indeed, the ODFs in Figure 15b,c show the well-known γ-fibre texture with the presence of different orientation components. The density orientations in 0A-FA (Figure 15b) are very weak compared to those in 1C-FA (Figure 15c). An examination of Table 5 indicates that the {554} <225> component, which is known to be an ideal component for deep drawing, is the most intense in 1C-FA. This texture could be produced by recrystallisation and derived from {332} <113>, the most beneficial transformation texture component for good deep drawability that also indicates good transition from hot to cold rolling after annealing. This last component is very intense in 1C-FA but is too weak in 0A-FA. Moreover, the orientations located in the γ-fibre, associated with the {111} <110> component, are also very high in 1C-FA but much less so in 0A-FA. These components, along with {554} <225>, favour the formability of the sheet. On the other hand, the {001} <110> and {112} <110> orientations located in the α-fibre, which worsen this formability, do not present a density of interest in either material.

## 4. Conclusions

The transformation texture of ferrite is influenced, amongst other processes, by the austenite to ferrite transformation, which leads to martensite. This is mainly associated with the 0A material, as shown by the phase diagrams and slab microstructures. As expected, given the rolling temperature, 0A-FM develops better ductility than 1C-FM, although the basic composition sheet develops a more pronounced anisotropy and lower deep drawability than the modified one in the final product.

Characterization by EBSD allows a more in-depth look into the microstructure evolution and texture development during the different production stages. The initial as-cast columnar is determined by the orientation {001} <100> cube, whereas the transformation of austenite to martensite is the origin of the texture {001} <110> rotated cube. Both components characterize the texture in the analysed materials, from the slab until the intermediate annealing stage, along with a microstructure characterized by a progressive decrease in martensite. During cold rolling, there is a transformation in the textures of both materials, and after final annealing the γ-fibre component dominates the texture in both materials. This component is much more intense in the 1C material and is supported by components that show good deep drawability ({554} <225>) and good transition from hot to cold rolling ({332} <113>).

The following conclusions can be drawn from the results obtained:

- During hot rolling, the materials use the storage energy to transform the martensite phase and to recrystallize the microstructure, with the latter mainly occurring in 1C

due to its less biphasic characteristics. Moreover, the presence biphasic regions in 0A maintains grain size and prevents them from overgrowing, as occurs in 1C after intermediate annealing.

- Due to the manufacturing specifications, after intermediate annealing the basic material develops a better recrystallization grade than the modified one.
- The stresses accumulated during cold rolling are used as energy in the final annealing, with 1C accumulating more energy as it is further reduced. In addition, the deformation energy in 0A allows the reorganization of elements, which results in the precipitation of carbides and nitrides, while in 1C the external distortion generates a strong dislocation grid that inhibits precipitation.
- The cube texture transforms to rotated cube, which then transform into {332} <113> during the phase transformation and shifts to {554} <225> during subsequent cold rolling; this latter component shows good deep drawability. Consequently, γ-fibre increases with processing in both materials, from hot rolling to final annealing. This fibre is strongly developed in the modified steel after final annealing, with an intense density of the components {332} <113> and {554} <225>.
- The Lankford coefficients and EBSD results confirm the evolution of texture throughout the manufacturing process for ferritic stainless steel, mainly in the material with modified composition, which exhibits a less pronounced anisotropy and higher deep drawability than the basic composition.
- As a result, the modified composition and process delivers a better recrystallisation status and, thus, better drawability performance.

**Author Contributions:** A.N.: Conceptualization, Investigation and Writing—original draft preparation; I.C.: Methodology, Resources and Formal Analysis; J.F.A.: Supervision and Validation; D.L.S.: Supervision and Writing—Reviewing and Editing. All authors have read and agreed to the published version of the manuscript.

**Funding:** This research was funded by the "Proyectos de I + D Individuales" programme, Centro para el Desarrollo Tecnológico Industrial (CDTI), Ministerio de Economía y Competitividad from the Spanish Government—project "FERRINOP".

**Data Availability Statement:** Not applicable.

**Acknowledgments:** Authors acknowledge the Program for the Promotion of Research and Transfer activity of the University of Cádiz, for its financial support.

**Conflicts of Interest:** The authors declare no conflict of interest.

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
