# Peer review of "Transformation of the Microstructure of Fe-Cr Steel during Its Production"

_metals, doi:10.3390/met11050806_

Round 1

Reviewer 1 Report

This manuscript describes the microstructure and texture changes during the manufacturing of Fe-Cr ferritic stainless steel. It presents some interesting results regarding the comparison of two different alloys, one with more austenite-stabilizing elements (basic) and the other with less C and N. While the content may be of interests to the industry, the manuscript can be improved. The English is somewhat difficult to understand. Here are some comments:

  1. Throughout the manuscript, there are numerous errors regarding the references to Figures and Tables. Please check the places where “Error! Reference source not found” was present.
  2. In the Abstract, when first mentioning g-fiber or a-fiber, the corresponding definitions should be given, i.e. <111>//ND and <110>//RD. Similarly, EBSD and ODF should also be given full names when they are introduced in the first time.
  3. In the Abstract, Lines 17-19, “The phase diagram, microstructure, Lankford coefficients and EBSD results confirm the evolution of texture during the manufactured processing of ferritic stainless steel”. The word “manufactured” should be deleted.
  4. In the Abstract, Lines 20-22, “Texture {001}<100> cube and {001}<110> rotated cube determine the orientation from the slab until the intermediate annealing stage”. This sentence should be written as “The cube ({001}<100>) and rotated cube ({001}<110>) textures dominate the crystal orientation from the slab until the intermediate annealing stage”.
  5. In Introduction, Lines 47-48, “Measurement of the orientations of polycrystalline grains (γ and α fibre) after material formability allows its deformation mechanism to be evaluated”, should be written as “Measurement of the orientations of polycrystalline grains (γ and α phases) after material deformation allows the evaluation of the deformation mechanism”.
  6. In Introduction, Lines 50-53, “This representation is done by way of Orientation Distribution Function (ODF) maps or diagrams, which are defined by Euler angles φ1, F, φ2 applied to the [100], [010] and [001] axes of the grain crystal structure, thus making them coincide with the RD, ND and TD (Transversal Direction) of the sample, respectively”. First, ODF is usually not called “maps”, but “sections”. Second, the Euler angles φ1, F, φ2 are defined as the consecutive rotations (angles) about the ND, the rotated RD and the rotated ND, respectively, that bring the sample coordinate system (RD, TD, ND) into coincidence with the crystal coordinate system ([100], [010] and [001]).
  7. Page 2, Lines 73-75, caption of Figure 1: “(a) Plan views of the φ2=45° section of Euler space (Bunge notation) applicable to bcc materials subjected to plain strain” should be “ (a) φ2=45° section of the Euler space …”. The words “… plain strain” should be “plane strain”. “(b) ODF function of material in study that represent typical γ and α texture positions.” Should be ”φ2=45° ODF section of the material in this study that shows typical γ and α fibers.”
  8. Page 4, Line 157, what is “coefficient of acrimony”? Line 163, what does “FM train” mean here?
  9. Page 5, Lines 174-175, “Salsa is the tool used to generate an ODF, an approach to texture measurements.” The word “measurements” should be “representation”.
  10. Page 6, Lines 212-213: “This transformation is greater the higher the content of austenite-stabilizing elements.” This sentence is not understandable. Do you mean “the higher the content of austenite-stabilizing elements, the greater the transformation”?
  11. Page 7, Line 251, “… from one sample to the next.” The word “next” should be “other”.
  12. Page 10, Line 334: “It can also be seend …” should be “It can also be seen …”.
  13. Page 11, Lines 369-370: “This suggests that most of the austenite was recrystallized to martensite by …”. Can “austenite” recrystallize to “martensite”? Wouldn’t it be “transformed”?
  14. Captions of Figures 9 and 11: “… crystallographic direction parallel to RD”. What do you mean here? Which crystallographic direction is parallel to RD? Also, a legend is needed to illustrate the color lines in the maps.

Reviewer 2 Report

Dear authors

The paper presented for review is generally well written, but before publication have to be corrected.

First, all figures are too small and because of that they are invisible and illegible.

Good would be to present chemical composition of both tested steels and very good would be to compare those values to the standard.

In table 1 what authors means by annealing time –short and long? This have to be explained.

In general nowadays should be use nm not Angstrem

Please use carefully microstructure and structure and do not mix those two things. Remember that if we are talking about grains and phases we are taking about microstructure. When we are talking about for example about crystallization as whole we are talking about structure. Structure is in macroscopic the point of view.

Do not be afraid of presenting microstructure in large images. Well-made are an ornament of article.

If authors present microstructures or macrostructures have to be write have they were etched.

Dear authors You are writing about simulation of rolling process. In such condition it must be presented parameters. At least temperature, strains and strains rate.

Good would be to present microstructure before band transformation and if authors have such microstructure views obtained with higher magnifications.

In general after correction the paper can be published but in this form leaves unsatisfied.

But after those correction in my opinion it will be a very good work.

Reviewer 3 Report

Dear Authors,

I have read your paper "Transformation of the microstructure of Fe-Cr steel during its production" carefully. 

This paper describes the investigation of the microstructure of Fe-Cr steel. The general novelty is the new composition of the steel. This stainless steel has good mechanical properties for manufacturing of the parts. 

The paper is easy to read.

Methods are properly described, so that other research groups may reproduce them.

The paper is interesting. However, it requires few corrections.

  1. Please, compare EN 1.4016 (in the abstract) and AISI 430 (in the text of the manuscript). I leave my suggestion to the authors for reflection.
  2. Please, add information about the processis themperatures in the table 3. 
  3. Please, add size marker for macrostructure (fig. 5).

 The paper can be accepted for publication after minor improvements.

Reviewer 4 Report

  1. Page 3, line 119, probably, it should be “EBSD” instead of “EBSP”? If otherwise, please, explain.
  2. Is it possible to provide more information about the rolling mill? How many stands, what is the length of rolls and their diameter, speed of rolling, regimes, etc.
  3. Could you, please, make the line for 0A-FM steel (Figure 6) with some darker color? In my variant of the figure, it is a bit difficult to identify it.
  4. Something is wrong with the references to figures and tables through all the text. I also suggest replacing names of the websites with the references (for instance, page 3, line 124).

I recommend major revision.
